# A New Sustainable PPT Coating Based on Recycled PET to Improve the Durability of Hydraulic Concrete

**DOI:** 10.3390/polym16091297

**Published:** 2024-05-06

**Authors:** Adrián Bórquez-Mendivil, Carlos Paulino Barrios-Durstewitz, Rosa Elba Núñez-Jáquez, Abel Hurtado-Macías, Jesús Eduardo Leal-Pérez, Joaquín Flores-Valenzuela, Blanca Alicia García-Grajeda, Francisca Guadalupe Cabrera-Covarrubias, José Miguel Mendivil-Escalante, Jorge Luis Almaral-Sánchez

**Affiliations:** 1Facultad de Ingeniería Mochis, Universidad Autónoma de Sinaloa, Ángel Flores S/N, Las Fuentes, Los Mochis 81223, Sinaloa, Mexico; adrian.borquez.fim@uas.edu.mx (A.B.-M.); durstewitz@uas.edu.mx (C.P.B.-D.); ronunez@uas.edu.mx (R.E.N.-J.); eduardo.leal@uas.edu.mx (J.E.L.-P.); jflores@uas.edu.mx (J.F.-V.); blanca.garcia@uas.edu.mx (B.A.G.-G.); guadalupe.cabrera@uas.edu.mx (F.G.C.-C.); 2Centro de Investigación en Materiales Avanzados, S.C., Miguel de Cervantes #120, Complejo Industrial Chihuahua, Chihuahua 31136, Mexico; abel.hurtado@cimav.edu.mx

**Keywords:** sustainability, PPT coating, recycled PET, durability, hydraulic concrete

## Abstract

A new, sustainable polypropylene terephthalate (PPT) coating was synthesized from recycled polyethylene terephthalate (PET) and applied onto a hydraulic concrete substrate to improve its durability. For the first step, PET bottle wastes were ground and depolymerized by glycolysis using propylene glycol (PG) in a vessel-type reactor (20–180 °C) to synthesize bis(2-hydroxypropyl)-terephthalate (BHPT), which was applied as a coating to one to three layers of hydraulic concrete substrate using the brushing technique and polymerized (150 °C for 15 h) to obtain PPT. PET, BHPT, and PPT were characterized by FT-IR, PET, and PPT using TGA, and the PPT coatings by SEM (thickness), ASTM-D3359-17 (adhesion), and water contact angle (wettability). The durability of hydraulic concrete coated with PPT was studied using resist chloride ion penetration (ASTM-C1202-17), carbonation depth at 28 days (RILEM-CPC-18), and the absorption water ratio (ASTM-C1585-20). The results demonstrated that the BHPT and PPT were synthetized (FT-IR), and PPT had a similar thermal behavior to PET (TGA); the PPT coatings had good adhesion to the substrate, with thicknesses of micrometric units. PPT coatings presented hydrophilic hydrophilic behavior like PET coatings, and the durability of hydraulic concrete coated with PPT (2–3 layers) improved (migration of chloride ions decreased, carbonation depth was negligible, and the absorption water ratio decreased).

## 1. Introduction

In recent decades, industrial growth and the expansion of the production of consumables have led to an exponential increase in the generation of polymeric waste, including polyethylene terephthalate (PET) [1]. PET is a thermoplastic polymer widely used in the production of beverage and food containers, which has reached a critical point in its life cycle, a situation that makes it an environmental and economic challenge due to its inadequate final disposal and permanence in the environment [2]. Faced with the urgent need to address the pollution generated by polymeric waste and the transition to more sustainable practices, PET recycling has become a fundamental response [3]. In 2019, an annual production of 30.5 million tons of PET was reported in the world, and by 2024, it is expected to be around 35.3 million tons [4], corresponding to the generation of PET bottles, which would subsequently be wasted, that has generated the interest in exploring alternatives for recycling and converting them into new materials with added value [5].

PET recycling can be carried out in different recycling phases: primary, secondary, tertiary, and quaternary. Mechanical and chemical methods have been commonly reported, which comprise the secondary and tertiary phases of their recycling, respectively [6]. Mechanical recycling consists of the collection, sorting, and grinding of PET bottles, resulting in the production of flakes, fibers, or powders [7]. On the other hand, chemical recycling, particularly by glycolysis, causes the controlled degradation of PET into monomers or oligomers of lower molecular weight, known as glycolyzed products [8]. These have been used for the synthesis of a wide range of materials, such as resins, coatings, and adhesives [9,10,11]. Powders obtained from post-consumer bottles (PET) were applied on mild steel substrates as anti-corrosive coatings [12]. However, it is important to consider that the PET bottle recycling process, especially when mechanical recycling is involved, is the most important generator of large quantities of microplastics. This is because, during the crushing and screening of PET bottles, to produce the sizes of this material and continue with the chemical process, very small-sized particles (microplastics) are also produced. These are collected in the wastewater generated by the process, and their treatment produces waste sludge in which said microplastics are trapped. At the time of their final disposal, they are transferred to the environment (through the sewage network, rivers, oceans, or soil), which becomes a challenge that should not be ignored [13]. Furthermore, humans are exposed to the ingestion of microplastics, which, if it becomes chronic, may represent a risk that affects the immune microenvironment [14].

In the construction industry, innovation has been fundamental to the development of sustainable and long-lasting solutions. Products derived from PET recycling have shown potential to improve the properties and useful lives of building materials. For example, PET reinforcing fibers have been applied for cementitious materials [15,16,17], PET powders for the manufacturing of asphalt mixtures [18], and PET resin for polymeric concrete [19]. Hydraulic concrete is the most used material in the construction industry, and maintaining its durability is very important to ensure that it meets the life service for which it was designed [20]. This can be achieved with proper mix design [21] or the application of coatings. Coatings of acrylic [22] and epoxy resin [22,23]; cementitious coatings modified with nanoSiO_2_ and hybrid nanoSiO_2_ [23]; silane cream (alkylalkoxysilane) and silane gel (triethoxysilane) [24]; acrylate terpolymer-based coatings [25]; magnesium fluorosilicate; waterglass; sodium fluorosilicate [26]; interior and exterior paints [27]; calcium silicate; and acrylic–silicon [28] have been applied to concrete and studied for carbonatation. Coatings of epoxy resin modified with graphene oxide [29], nano-silica, silane/nanoclay [30], chlorinated rubber, polyurethane [31], and alkiltrialkoxisilane [32] have been used on concrete to improve its resistance and permeability to chloride ions. Coatings of epoxy resin modified with graphene oxide [29], silane-based water repellent agents such as alkylalkoxy silane [33], isobutyltriethoxysilane [34], fluoroalkyl silane modified with rice husk ash [35], and silane/siloxane [32] have been employed on concrete and their performance in water absorption has been studied. Hydraulic concrete coated with polyurethane has presented a low risk of corrosion by obtaining electrical resistivity [36].

After an exhaustive literature review, studies on the synthesis and application of PPT coatings based on recycled PET to improve the durability of hydraulic concrete were not found. The success of our research lies in developing a new, sustainable polypropylene terephthalate (PPT) coating from recycled polyethylene terephthalate (PET), synthesized using a simple method and applied with a brush, which improved the durability of hydraulic concrete. BHPT and PPT were characterized by FT-IR, and PPT and PET by TGA. The coatings were characterized by SEM, contact angle, adhesion, carbonation depth, permeability of chloride ions, and water absorption.

## 2. Materials and Methods

### 2.1. Materials

Post-consumer PET soft drink bottles were mechanically recycled by grinding to reduce them to flakes of 1/2 in, then washed with a 10% NaOH solution. Propylene glycol (PG) ≥ 99%, zinc acetate (ZnA), tetrahydrofuran (THF), ethanol, and distilled water were used for the depolymerization of PET and to obtain BHPT. The reagents were supplied by Sigma-Aldrich, Toluca, México. Ordinary Portland cement type III, from the CEMEX brand, CPC, according to ASTM C150/C150M-22 [37], as well as natural aggregates (gravel and sand), were purchased from CONSTRURAMA, S.A. of C.V., Los Mochis, Sinaloa, México. Water was supplied by the local drinking water network.

### 2.2. Synthesis of the Bis(2-hidroxyproyl) Terephthalate (BHPT) Monomers

The chemical recycling process for PET bottles involved their depolymerization with propylene glycol to obtain the BHPT monomers. The procedure was carried out in the following stages: The PET flakes were deposited in a Syrris vessel-type reactor CEIMX-20110606-GLOBE at room temperature, this equipment was supplied by the Sirrys company located in Royston, United Kingdom, and is installed at the Facultad de Ingeniería Mochis de la Universidad Autónoma de Sinaloa, México. PG was added at a rate of 100% according to the weight of the PET, and zinc acetate was added as a catalyst at a rate of 0.5% according to the weight of the PET. Afterward, these reagents were mixed by means of mechanical stirring at room temperature for 15 min. Next, the temperature increased at a rate of 5 °C/min from 20 °C to 180 °C for 3 h. Then, it was kept at rest until it reached room temperature for later use. As a result of the process, the product was likely to be BHPT, and excess unreacted PG and EG were formed as by-products.

### 2.3. Purification Process of BHPT

The BHPT monomers obtained were subjected to a purification process to eliminate excess unreacted PG and EG formed as by-products during synthesis. EtOH and THF, 100% and 20% by volume of BHPT, respectively, were stirred in a beaker for 15 min. Then, the BHPT monomers were added to the mixture and stirred for 30 min at room temperature. Next, three washes were performed with distilled water, each with a volume equivalent to 30%, according to the volume of BHPT for 15 min. The solution was vacuum-filtered in a Buchner funnel, and after that, it was dried in an oven with air circulation at 60 °C for 24 h and kept at rest until reaching room temperature for use in later stages. 

### 2.4. Synthesis of Polypropylene Terephthalate (PPT)

The synthesis of PPT was carried out in the following steps: (1) The purified BHPT monomer, at room temperature, was poured into circular molds 2.5 cm in diameter and 1.5 cm in height to prepare bulk composites. Afterward, (2) the molds with the BHPT were placed in an oven with air circulation at 150 °C for 15 h for polymerization, and thus, the PPT was synthesized.

### 2.5. Elaboration of Cementitious Concrete Mixes

A low-strength concrete mixture (150 kg/cm^2^) with a water/cement ratio of 0.8 was obtained, with the objective of studying the effect of the polypropylene terephthalate (PPT) coating on porous concrete. Table 1 shows the dosage used, which was calculated following the method established by the ACI 211.1-22 [38]. The preparation of the mixture was carried out as described in the ASTM C31/C31M-22 standard [39]. This mixture was poured into cylindrical metal molds, which were 10 cm in diameter and 20 cm high. Once the mixture had set, it was cured in water for 28 days. After this period, it was cut into slices 5 cm thick and stored for the later application of the coating.

### 2.6. Elaboration of PPT Coatings Hydraulic Concrete Using a Brush

The hydraulic concrete specimens had previously been prepared to ensure adequate application and adhesion of the PPT coating with its surface, which was free of contaminants, laitance, loose concrete, and dust, in accordance with the SSPC-SP-13/NACE No. 6 standard [40]. Subsequently, the BHPT was applied to the surface of the concrete using a brush in three layers at room temperature, and for each coated layer, the specimens were placed in an oven with air circulation under the polymerization conditions described in 2.4 to subsequently achieve PPT coating formation. The nomenclature for the studied specimens was declared as follows: REF for uncoated reference concrete and CB1, CB2, and CB3 for concrete coated with 1, 2, and 3 layers, respectively.

### 2.7. Characterization

#### 2.7.1. FT-IR

The PET, PPT, and BHPT samples were analyzed using the FT-IR technique to identify their characteristic bonds using a Nicolet iS50 spectrometer with transmission configuration, supplied by Thermo Fisher Scientific México; Monterrey, Nuevo León, México, and is installed at the Centro de Investigación en Materiales Avanzados, S.C., Chihuahua, México (CIMAV). The three types of materials had different treatments for FT-IR analysis. PET was reduced to powder; PPT was polymerized (in oven at 150 °C for 15 h) and reduced to powder; and BHPT was analyzed in a liquid state. The samples were placed on a diamond surface, and spectroscopic measurements were carried out within the wavelength range of 4000 to 400 cm^−1^ with the help of attenuated total reflectance fixtures.

#### 2.7.2. TGA

The thermal stability of PET and PPT was evaluated by thermogravimetric analysis (TGA). This analysis was performed using the TA Instrument SDT-Q600 simultaneous equipment under an oxygen atmosphere, with a heating rate set at 10 °C per minute, within a temperature range spanning 0 to 700 °C, supplied by Waters S.A. de C.V.; Benito Juárez, Ciudad de México, México, and is installed at the CIMAV.

#### 2.7.3. Physical Properties of Coatings

The thickness of the coating was examined with a scanning electron microscope (SEM); this analysis was performed with the JEOL SEM model JSM5800LV microscope, supplied by JEOL de Mexico S.A. de C.V.; Benito Juárez, Ciudad de México, México, and is installed at the CIMAV. The samples consisted of small representative fragments of coated concrete with three layers of PPT, without prior preparation to avoid modification or degradation of the organic coating.

The adhesion of the coatings was evaluated using Cross Section Test Method B of ASTM D3359-17 [41], in which a grid is made on the coated substrate with a line spacing of 2 mm, and then adhesive tape is placed over the grid using pressure and removed after a certain amount of time. Adhesion strength grades are based on the percentage of the coating removed after testing, with levels ranging from 0B (very poor adhesion) to 5B (very good adhesion).

The wettability of coatings, in order to determine their hydrophilicity or hydrophobicity, was determined using an FTA 200 contact angle analyzer (First Ten Armstrong, Portsmouth, VA, USA) with samples of the coated substrates (3 cm × 9 cm), which were placed horizontally. Drops of 10 µL of distilled water were applied six times to the surface of the coating, and the resulting data were analyzed to calculate the mean value and standard deviation, which were then reported.

#### 2.7.4. Durability Tests

The water absorption rate was evaluated according to ASTM C1585-20 [42], in which the evaluated samples consisted of slices 10 cm in diameter and 5 cm high, in which only one side was exposed to a sheet of water of 2 ± 1 mm at times of 1, 5, 10, 20, and 30 min and 1, 2, 3, 4, 5, and 6 h. Afterwards, the absorption produced by capillarity was calculated, in mm, according to the mass difference.

The resist chloride ion penetration was carried out according to the ASTM C1202-19 [43] standard, in which the samples consist of slices 10 cm in diameter and 5 cm high with only one face covered with the coating. The samples were placed between two cells: one acted as a cathode electrode filled with NaOH aqueous solution (0.3 N) in which the uncovered face was exposed, and the other one as anode electrode filled with NaCl aqueous solution (3%) in which the covered face was exposed. The chloride ion permeability classification of the specimens was obtained in accordance with the same standard. 

The carbonation depth was assessed according to the RILEM CPC-18 standard [44], after 7, 14, and 28 days of exposure to CO_2_ in an accelerated carbonation chamber with the following conditions: temperature of 25 ± 3 °C, average relative humidity of 60 ± 5%, and average CO_2_ concentration of 5%. The specimens were sprayed with a diluted phenolphthalein solution (1% in ethyl alcohol); this is an indicator that reacts with non-carbonated areas, changing the color to purple. The carbonation depths of the samples were obtained with an average of 9 measurements made with a digital vernier.

## 3. Results and Discussion

### 3.1. Proposed Schematic Drawing of the Synthesis of PPT from PET Recycling

Figure 1 shows a proposed schematic drawing of the synthesis of polypropylene terephthalate (PPT) from the chemical recycling of PET wastes. PPT is synthesized in two stages: First, (1) Depolymerization of PET by glycolysis using PG as a solvent with catalyst to form BHPT monomers, and excess unreacted PG and EG are formed as by-products. After that, the BHPT monomers are purified to remove excess PG and EG [8]. In a second stage, (2) Polytransesterification of BHPT monomers under temperature conditions to form PPT [45,46].

### 3.2. FT-IR

Figure 2 presents the FT-IR spectra of the PET, BHPT, and PPT samples: (a) shows the entire range of wave numbers studied, and (b)–(d) are amplified regions of study. The spectra, previously normalized to their maximum transmittance values, were arbitrarily shifted on the same transmittance axis. In (a), the following bands were observed: For PET, C=O stretching bands (carbonyl group) around 1720–1750 cm^−1^, asymmetric and symmetric C-H stretching bands (CH_2_-group) around 2950–2960 and 2880–2900 cm^−1^, and C=C stretching bands in the aromatic ring around 1600–1620 cm^−1^ were identified [47]. For BHPT, the same characteristic bands as in PET were observed, which confirmed the persistence of the basic structure of PET (1720–1750 cm^−1^ for C=O stretching, 2950–2960 and 2880–2900 cm^−1^ for CH_2_ and CH groups, respectively, and 1600–1620 cm^−1^ for C=C stretching bands). Furthermore, a new broad band was identified between 3200 and 3600 cm^−1^, corresponding to the stretching of the hydroxyl group (OH) [48], and a band around 2980 cm^−1^, attributed to the presence of CH_3_ groups, confirmed that BHPT was obtained by transesterification between PET and propylene glycol [49]. The spectrum of PPT is visually very similar to that of PET, with some differences related to the presence of methyl groups in its molecules produced during the polytransesterification of BHPT. These differences can be observed in detail in the magnifications presented in (b), (c), and (d), which are shown below. In (b), the magnification of the C-H stretching region from 3020 to 2860 cm^−1^) is shown; It is observed that the PET bands (indicated by black dashed lines) at 2968 cm^−1^ and 2908 cm^−1^ remained with a shift toward lower wavenumbers in PPT (indicated by red dashed lines), at 2952 and 2883 cm^−1^, respectively. Furthermore, a band at 2981 cm^−1^ was observed in the PPT spectrum (indicated by a red dash line), which was absent in the PET spectrum., and this difference was attributed to the presence of methyl groups in the PPT molecules. In (c), the magnification of the PET and PPT spectra region from 1500 to 1350 cm^−1^, where We can observe in PPT the appearance of two bands at 1452 and 1379 cm^−1^ (indicated by red dashed lines), corresponding to bending vibrations of the CH_3_ groups, signals absent in the PET spectrum. In (d), the magnification of the PET and PPT spectra region from 1000 to 800 cm^−1^. In PPT, the bands at 987, 937, 920, and 811 cm^−1^ are observed (indicated by red dashed lines), which are characteristic of polypropylene [50,51]. 

From the above, it can be concluded that the differences shown between the PET and PPT spectra are due to the fact that PET was synthesized from ethylene glycol, while PPT was synthesized from propylene glycol.

### 3.3. TGA

Figure 3 shows the TGA of PET and PPT. For PET, a two-stage degradation was observed: The first mass loss was of 93% (361–450 °C), related to the random cleavage of the ester bond resulting in the formation of oligomers, and the second degradation observed was of 7% (450–518 °C), attributed to carbonyl group decomposition [52,53,54,55,56,57]. For the PPT, the first mass loss was of 2.5% (30–210 °C), related to residual -OH groups that did not participate during the polytransesterification of the BHPT, and a second mass loss of 10.5% (210–312 °C) occurred, attributed to the decomposition of the CH group. A third mass loss of 80% (312–445 °C) was also observed, which may have corresponded to the presence of a carbon–carbon bond that promoted the random cleavage of the ester bond mechanism with increasing temperature [52]. The results of TGA show that the main degradation of the PPT sample started at a temperature near (312 °C) that of PET (348 °C). That indicates that PPT has thermal behavior similar to the PET.

### 3.4. Physical Properties of PPT and Coatings

#### 3.4.1. Appearance

Figure 4 shows the appearance of the PPT obtained through the polymerization of BHPT. A reddish-brown solid bulk sample capable of being demolded is observed.

Figure 5 shows photographs of the reference concrete specimens (REF) and specimens coated with one layer (BC1), two layers (BC2), and three layers (BC3) of PPT. It can be seen that in BC1, the shade of the mortar paste did not present a visible change, so it was similar to that of REF. In BC2, most of the section became darker in REF, while in BC3, the entire section was darker and more uniform in color than BC2. At first glance, it can be concluded that, in BC2 and BC3, the coatings were perceptible, and that in BC3, it was applied to the entire surface.

#### 3.4.2. Thickness

Figure 6 shows the SEM micrograph of the cross section of coated concrete with three layers of PPT, with several thickness measurements, which were very similar. The average thickness of the PPT coatings was 2.50 μm.

#### 3.4.3. Adhesion

Figure 7 shows three photographs of samples BC1, BC2, and BC3, which were tested using ASTM D3359-17 (cross section testing method). It can see that there was no detachment of material from the interior of the squares formed when making the cuts (area removed from 0%). Therefore, the samples presented very good adhesion with the concrete, and were classified as 5B according to the cited standard. These results are similar to those reported for coatings on glass and metal substrates, made with resins synthesized from glycolized PET products [9,11].

#### 3.4.4. Wettability Capacity (Water Contact Angle, WCA)

The average WCA of the PPT coating was 72.90° ± 1.51. This measurement indicates hydrophilic behavior similar to that reported for PET sheets (78°) [58], PET films (60°), PET/fluorinated oligomeric polyester composite coatings (70–88°) [59], and PET coatings (68–78°) [60].

### 3.5. Durability Properties of Concrete

#### 3.5.1. Rate of Absorption of Water (ASTM C1585-20)

Figure 8 shows the rates of absorption of water of REF, BC1, BC2, and BC3, where REF-BC1 and BC2-BC3 overlap, respectively. REF and BC1 had the highest rates of absorption of water, which indicates that BC1 had the same water absorption rate as REF. It can be inferred that the BC1 coating behaved like an impregnation; it was absorbed by the concrete and did not present resistance to water absorption by capillarity. BC2 and BC3 had the lowest rates of absorption of water, and it can be inferred that the BC2 coating formed a layer with a sufficient water absorption rate compared to BC3, so one more layer (BC3) would not be necessary in order to improve that property.

#### 3.5.2. Permeability of Ion Chloride

Figure 9 presents the charge passed results for REF, BC1, BC2, and BC3. The samples presented average charge passed values of 7520, 5958, 2030, and 1775 C for REF, BC1, BC2, and BC3, respectively. According to the classification established by the ASTM C1202-19 standard, REF and BC1 had high permeability (>4000 C), BC2 had moderate (2000–4000 C) to low permeability (1000–2000 C), and BC3 had low permeability. A tendency of the permeability of the samples to decrease as the number of coating layers increased was observed. BC3 indicated that it had a greater ability to reduce the movement of chloride ions through concrete, but BC2 also showed a similar behavior to BC3, because its permeability was in the lower-to-moderate limit range and very close to the upper limit of low permeability.

#### 3.5.3. Electrical Resistivity

Figure 10 presents the average electrical resistivity results for REF, BC1, BC2, and BC3. The average electrical resistivity results were 8.98, 10.24, 23.53, and 48.84 kΩ·cm for REF, BC1, BC2, and BC3, respectively. According to the CEB-192 standard [61], reinforced concrete with electrical resistivity less than 10 kΩ·cm has a high risk of corrosion. The results showed that the risk of concrete corrosion decreased as the number of PPT coating layers increased. BC2 and BC3 showed a low risk of corrosion.

#### 3.5.4. Carbonation Depth

Figure 11 shows the results of carbonation depth for REF, BC1, BC2, and BC3 at 7, 14, and 28 days of exposure. For the REF and BC1 samples, it can be observed that, for 7 and 14 days, a violet color appeared, characteristic of the reaction of phenolphthalein with the non-carbonated zone, and for 28 days, this color disappeared, which indicates that the sample carbonated. In general, these two samples showed progressive development of carbonation until it was completely produced within a period of 28 days. In the samples BC2 and BC3, it can be observed that, for 7, 14, and 28 days a violet color appeared, which indicates that neither sample carbonated during that period.

Table 2 shows the carbonation depth measurements for REF, BC1, BC2, and BC3. The values for REF and BC1 were very similar, which indicates that one coating layer was not enough to offer protection to the concrete against CO_2_. The measures for BC2 and BC3 were the same. These coatings served as a protective barrier against CO_2_, which may be due to the fact that the chemical composition of the material with which the concrete was coated was similar to that of PET. This is why the material was able to have the characteristic of being chemically inert in this case, with the accelerated CO_2_ environment [62,63].

## 4. Conclusions

A new, brush-applied PPT coating on hydraulic concrete substrate was synthesized by polytransesterification of BHPT obtained from recycled PET bottles.

A proposed reaction mechanism for the synthesis of PPT from the glycolysis of PET was established.

The transformation of PET to PPT was achieved, as confirmed by FT-IR, because the incorporation of methyl groups in the PET molecule was verified by the appearance of CH_3_ groups and a significant reduction in OH.

The PPT coating had a hydrophilic behavior similar to that of PET (contact angle), and optimal adhesion (5B, ASTM D3359-17) similar to resins synthesized from glycolyzed PET products. It had a thermal behavior similar to that of PET, with initial main degradation at a temperature close (312 °C) to that of PET (348 °C).

The PPT coating based on recycled PET improved the durability of hydraulic concrete. The two-layer coatings had good performance in terms of protection from water absorption and carbonation; however, it was necessary to apply three layers in order to also protect against the penetration of chloride ions and the risk of corrosion. Thus, the use of three layers of PPT (with an average thickness of 2.50 μm) is recommended to protect the coating in the four properties mentioned.

## Figures and Tables

**Figure 1 polymers-16-01297-f001:**
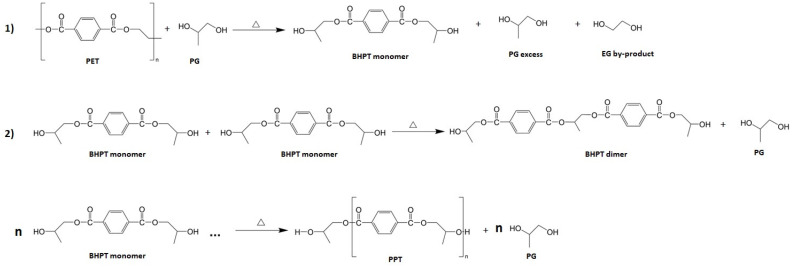
Proposed schematic drawing of the synthesis of PPT from PET recycling. (1) Depolymerization of PET by glycolysis using PG as a solvent with catalyst to form BHPT monomers. (2) Polytransesterification of BHPT monomers under temperature conditions to form PPT.

**Figure 2 polymers-16-01297-f002:**
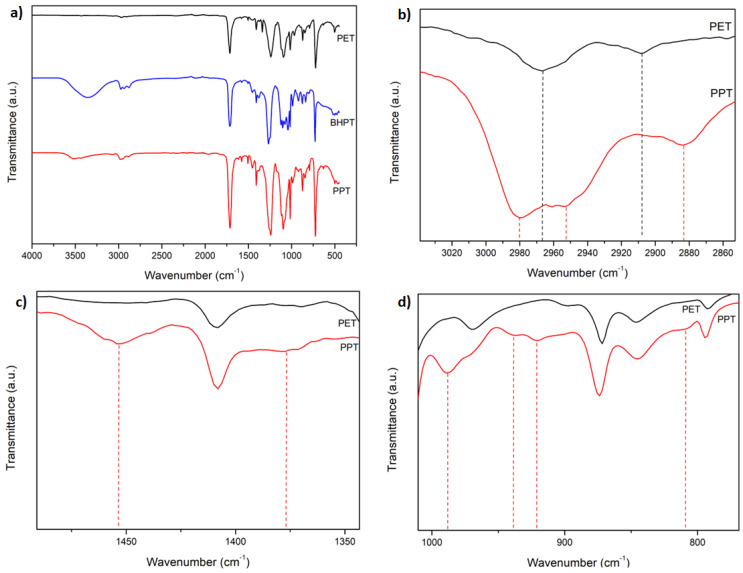
FT-IR spectrum of: (**a**) PET, BHPT, and PPT at a wavenumber region of 4000–400 cm^−1^, (**b**) C-H stretching region (3020 to 2860 cm^−1^), the PET bands (black dashed lines) with a shift toward lower wavenumbers in PPT (red dashed lines) (**c**) The magnification of the PET and PPT spectra region from 1500 to 1350 cm^−1^, where can observed in PPT the appearance of two bands (indicated by red dashed lines), of the CH_3_ groups, absent in the PET spectrum., and (**d**) the magnification of the PET and PPT spectra region from 1000 to 800 cm^−1^. In PPT, the bands are observed (indicated by red dashed lines), which are characteristic of polypropylene.

**Figure 3 polymers-16-01297-f003:**
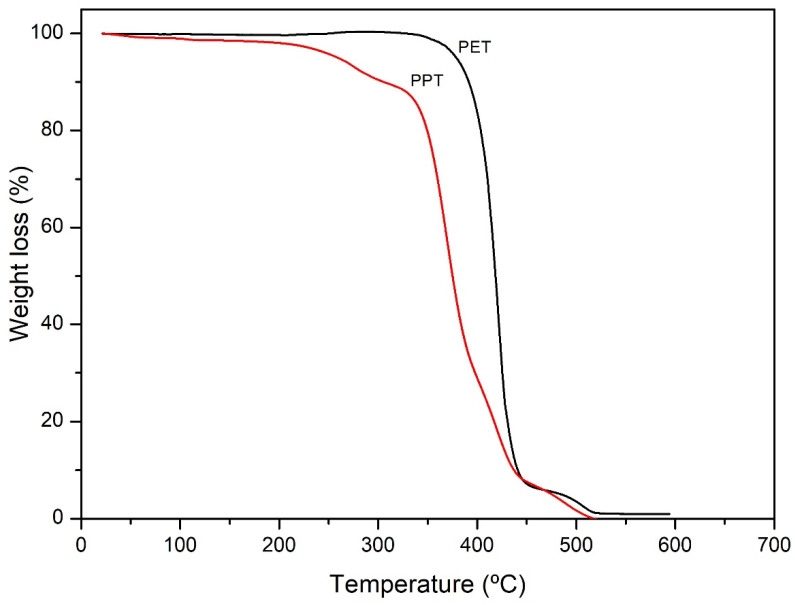
Thermogram of PET and PPT samples.

**Figure 4 polymers-16-01297-f004:**
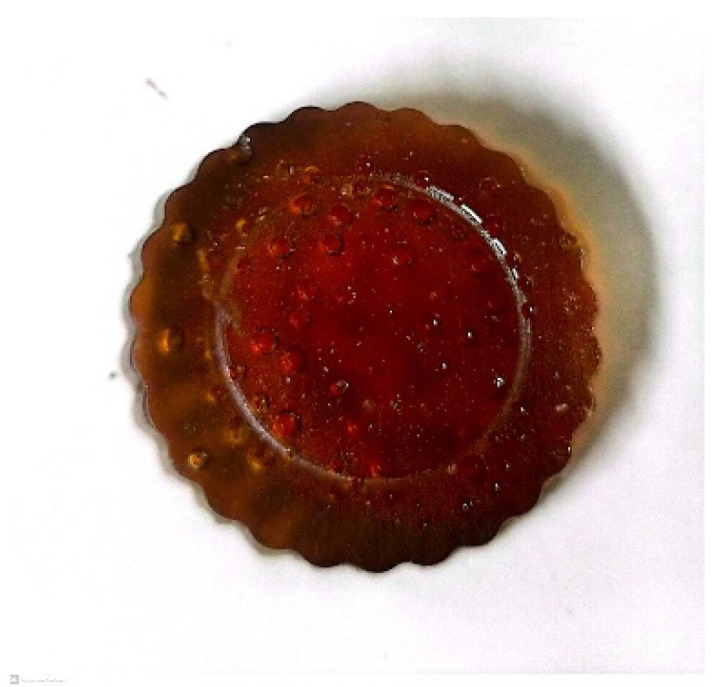
Appearance of PPT bulk sample.

**Figure 5 polymers-16-01297-f005:**
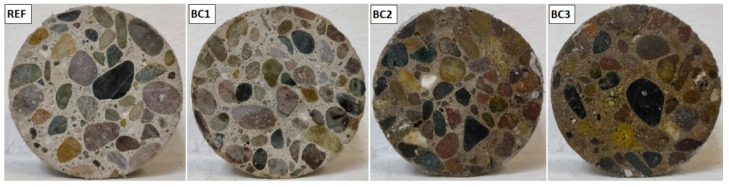
Appearance of the uncoated concrete specimens (REF) and those coated with PPT (BC1, BC2, and BC3).

**Figure 6 polymers-16-01297-f006:**
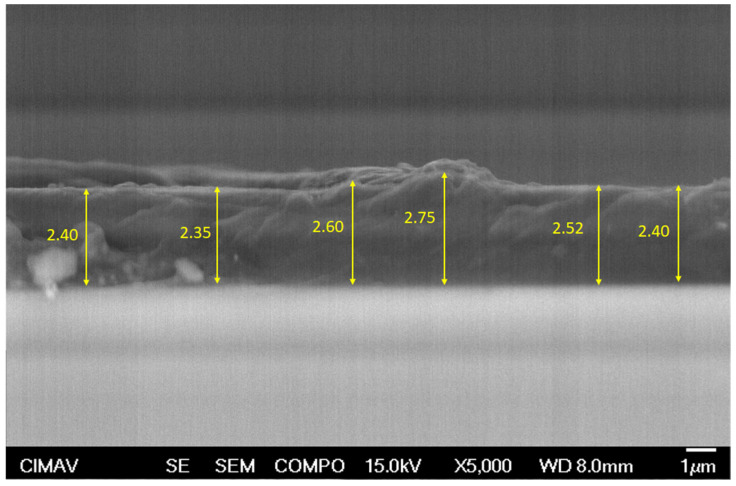
SEM micrograph of the cross section of a PPT coating with several thickness values.

**Figure 7 polymers-16-01297-f007:**
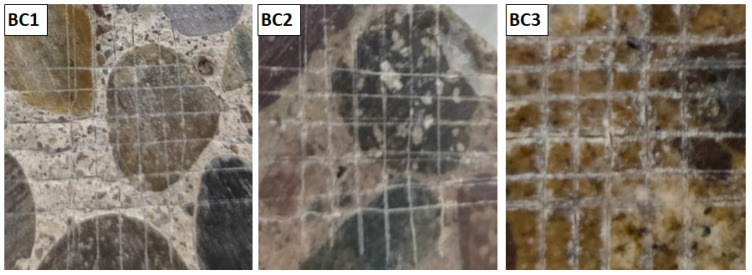
Photographs of the samples (BC1, BC2, and BC3) tested by cross-cut method adhesion (ASTM D3359-17).

**Figure 8 polymers-16-01297-f008:**
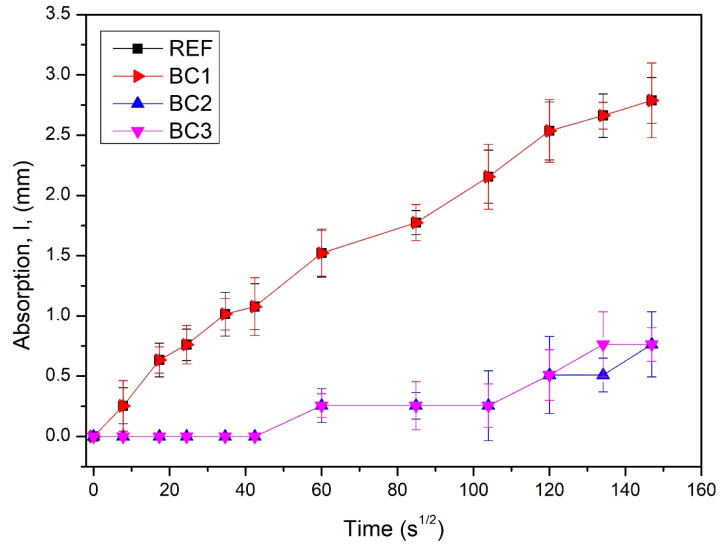
Water absorption rate graph of the uncoated (REF) and coated concrete specimens (BC1, BC2, and BC3).

**Figure 9 polymers-16-01297-f009:**
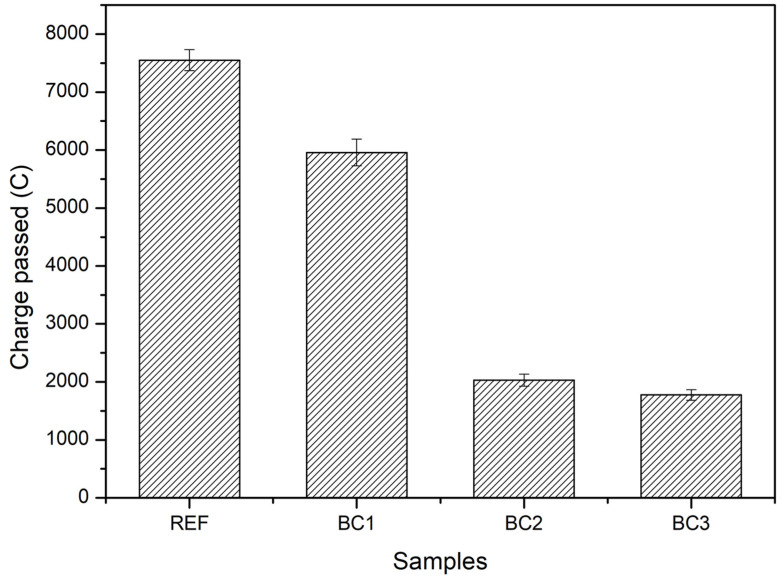
Charge passed to chloride ion in the uncoated (REF) and coated concrete specimens (BC1, BC2, and BC3).

**Figure 10 polymers-16-01297-f010:**
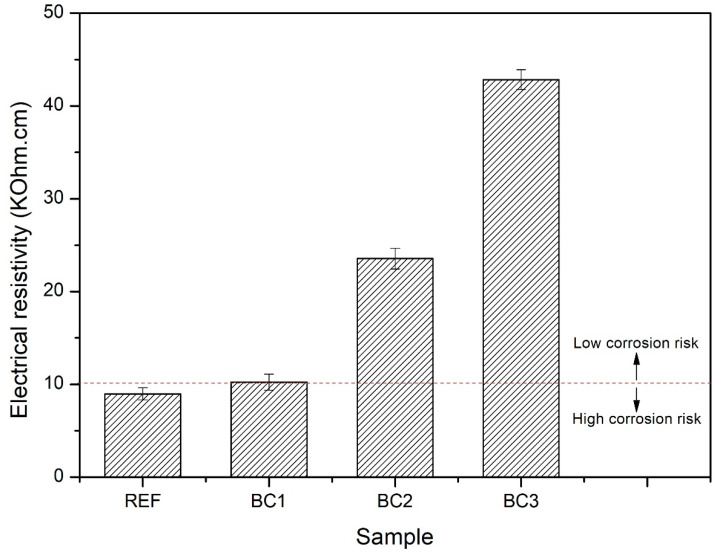
Electrical resistivity of the uncoated (REF) and coated concrete specimens (BC1, BC2, and BC3).

**Figure 11 polymers-16-01297-f011:**
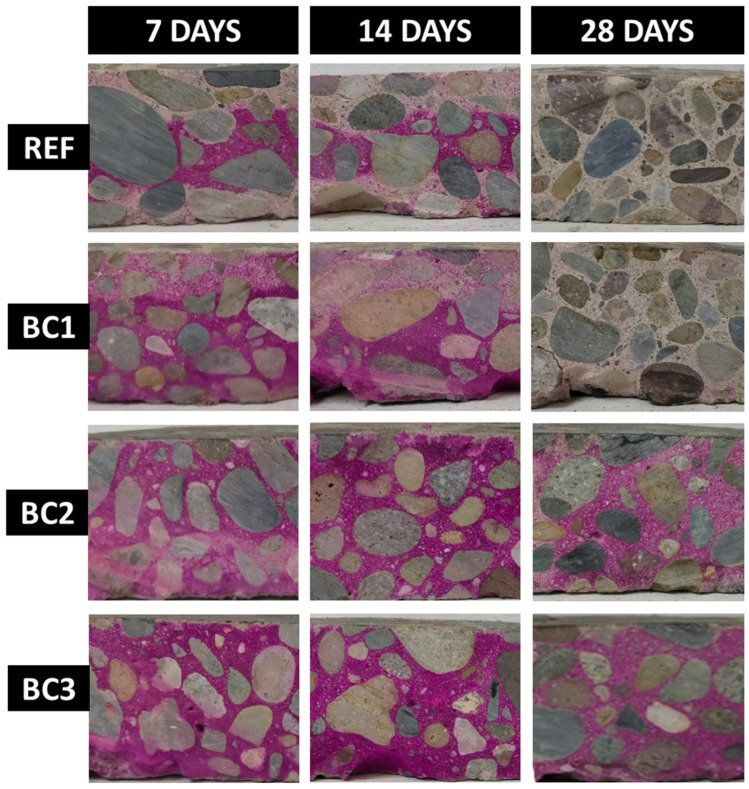
Carbonation depth of uncoated (REF) and coated concrete samples (BC1, BC2, and BC3) at 7, 14, and 28 days of CO_2_ exposure.

**Table 1 polymers-16-01297-t001:** Dosage of the mixture for 1 m^3^ of hydraulic concrete.

Material	Dosage (Kg/m^3^)
Cement	256.25
Water	205.00
Coarse aggregate	1024.00
Fine aggregate	608.40

**Table 2 polymers-16-01297-t002:** Carbonation depth measurements for uncoated (REF) and coated concrete specimens (BC1, BC2, and BC3) at 7, 14, and 28 days of CO_2_ exposure.

Carbonation Depth (mm)				
Exposure Time/Samples	REF	BC1	BC2	BC3
7 days	14.97	11.27	0	0
14 days	15.65	15.65	0	0
28 days	25	25	0	0

## Data Availability

Data are contained within the article.

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
