# Peer review of "A New Sustainable PPT Coating Based on Recycled PET to Improve the Durability of Hydraulic Concrete"

_polymers, 2024, doi:10.3390/polym16091297_

Round 1

Reviewer 1 Report (Previous Reviewer 1)

Comments and Suggestions for Authors

The authors have done significant work and have improved the quality of presentation of the study.
The only drawback, from my point of view, remains the proposed reaction mechanism.
The mechanism for producing polypropylene terephthalate from BHPT is polytransesterification, as on page 418 [42]. For the convenience of the authors, I am attaching a mechanism I converted from their work.
If the authors insist on using their proposed mechanism (polyesterification), the following data can support it:
1. A noticeable peak of C-O-C bonds in the IR spectrum;
2. Data on the composition of the mixture of low molecular weight compounds released during polycondensation (according to the proposed mechanism, they should not contain 1,2-propylene glycol).
Still, I recommend reconsidering the proposed mechanism as it does not meet theoretical expectations.

Author Response

Thank you very much for taking the time to review this manuscript. Please find detailed responses in the attached file and corresponding revisions/corrections following changes to the forwarded files.

Reviewer 2 Report (Previous Reviewer 2)

Comments and Suggestions for Authors

The manuscript has been improved following suggestions made in the first review round. I consider that authors take all the recommendations made for reviewers and they transform the original manuscript into a publishable work in Polymers Journal.

Author Response

Thank you very much for taking the time to review this manuscript. Please find detailed responses in the attached file and corresponding revisions/corrections following changes to the forwarded files.

Round 2

Reviewer 1 Report (Previous Reviewer 1)

Comments and Suggestions for Authors

Thank you for your thorough approach to my recommendations! From my point of view, the article is completely ready for publication.

This manuscript is a resubmission of an earlier submission. The following is a list of the peer review reports and author responses from that submission.

Round 1

Reviewer 1 Report

Comments and Suggestions for Authors

The work represents a broad applied research in the field of PET recycling and the use of its products. The quality of the graphic materials in the article is very high. Also, the properties of coatings and concrete have been studied exhaustively and in great detail. Unfortunately, I am forced to recommend rejecting the work, since it has significant shortcomings in terms of chemical processes, primarily the glycolysis of PET with propylene glycol. However, I strongly recommend resubmitting the article after making corrections.

44-46: I recommend adding more links to review articles on PET recycling methods (for example, 10.3390/membranes12111105).

52: The correct format for the link would be [8-10]. The similar correction is needed later in the text.

108: Sections 2.3-2.4 are missing. Probably description of PPT synthesis is absent.

184-185: Correct "BETH" to "BHET" please.

185-186: I recommend removing the designation “mmHg” for PET synthesis (fig. 2), and also clarifying in the text that “established pressure conditions” represent vacuum, not increased pressure.

191: The proposed mechanism in Figure 2 nicely captures the key shortcomings of the paper.

Firstly, the interaction of PET and PG will not produce the monomer shown. Ethylene glycol has a higher boiling point than propylene glycol, and both will undoubtedly remain in the reaction mixture. To obtain BHPT, it is likely necessary to introduce a much larger volume of PG throughout the entire process, gradually distilling the mixture of EG and PG from the reaction mixture. As a result of the process described in paragraph 2.2., the product will be a mixture of BHET, BHPT and 2-hydroxyethyl 2-hydroxypropyl terephthalate.

Secondly, the product labeled "BHPT oligomer" is not olygopropyl terephthalate. The reaction shown is a side reaction and undoubtedly occurs, but unlikely to a significant extent.

The same goes for this product in the third reaction: if you wanted to show this side reaction, why aren't this link shown in the PPT?

196-197: Figure 3 makes no sense in a scientific paper. I recommend removing it or using it as part of a graphic annotation.

199-288: Without a calibration curve, the IR spectrum cannot unambiguously confirm that the specified substance is BHPT or PPT, it only indicates the presence of certain functional groups. I recommend obtaining pure standard BHPT and PPT from terephthalic acid or dimethyl terephthalate and 1,2-propylene glycol, then taking their spectra and comparing them with those indicated.

229-243: I recommend placing thermal properties in a separate section from FTIR.

Reviewer 2 Report

Comments and Suggestions for Authors

The manuscript entitled A New Sustainable PPT Coating Based on Recycled PET to Improve the Durability of Hydraulic Concrete" from A. Bórquez-Mendivil et al. presents an interesting approach for recycling polyethylene terephthalate from waste bottles to be used as a coating for protecting the concrete surface. Although the approach presented by the authors is very interesting, the manuscript presents some issues that must be corrected before being accepted for publication in Polymers.

1) The introduction is adequate for this manuscript, although the authors should point out the possibility of microplastic pollution after the application of PPT coating.

2) I consider that Figure 1 should be part of the results and discussion instead of materials and methods

3) The mechanism proposed in Figure 2 isn´t a mechanism because this figure does not have arrows to indicate electron movements. The same observation applies to Figure 3: it is not necessary to point out that PET has a different chemical structure from PPT

4) In Figure 6 it is shown a SEM micrograph of a single sample showing a cross-section of concrete coated with PPT. However, the authors point out in the material section that they prepared three different samples containing one, two, and three layers. I suggest that the authors show all the SEM images of these samples (BC1, BC2, BC3) and also show the SEM images of reference sample (uncoated)

5) Figure 8 is just using space in the manuscript. Considering that the authors are informing only the contact angle of one sample, this figure should be removed. The contact angle section must be improved, removing Table 2 and including these references in the discussion.

6) What is the charge passed? This concept is shown in Figure 10. I think that this was a bad Spanish-english translation. Please revise and modify if necessary.

Comments on the Quality of English Language

The quality of English language must be improved. It looks like a literal spanish to english translation.